# What is the best proxy for political knowledge in surveys?

**Lauri Rapeli** [ID] *

The Social Science Research Institute, Åbo Akademi University, Turku, Finland

* lauri.rapeli@abo.fi

## Abstract

Online surveys are becoming the dominant form for survey data collection. This presents a problem for the measurement of political knowledge, because, according to recent scholarship, unsupervised measurement of political knowledge in web-based surveys suffers from respondent dishonesty. This study examines the validity of five possible survey proxies for political knowledge: self-assessed sophistication, political interest, internal political efficacy, accuracy of party placements on a left-right dimension and political participation. The analysis draws on a 2020 survey data (n = 1,097) and partial replications with identical measures from a 2008 survey data (n = 1,021) from Finland. Through several tests, the five proxies are assessed in terms of convergent validity, criterion validity and predictive validity. Across all tests, political interest performs best on all dimensions of validity and demonstrates largely identical relationships with political knowledge. Although the survey measurement of political interest and political knowledge may partly tap into slightly different constructs, the analysis supports the conclusion that political interest is the most suitable survey proxy for political knowledge from among the five proxy candidates included in the analysis.

**Data Availability Statement:** The Finnish Social Science Data Archive. The 2008 data (FSD2499 Knowledge of Politics and Society 2008): https://services.fsd.tuni.fi/catalogue/FSD2499?study_language=en&lang=en. The 2020 data has been deposited and will be available in late 2023. Until

## Introduction

Political knowledge, often defined as 'the range of factual information about politics that is stored in long-term memory' [1], belongs to the most commonly deployed variables in the study of political behavior. Political knowledge is the prime empirical indicator of political sophistication, i.e., expertise in the political domain [2]. Used both as a dependent and an independent variable, political knowledge is widely considered as an important facilitator of meaningful political participation. Its analytical significance is based on a widespread scholarly consensus, according to which an enlightened citizenry is a crucial condition for the functioning of representative democracy [1].

Scholars (almost exclusively) rely on survey data for the measurement of political knowledge. The preferred method is asking factual knowledge questions, i.e., questions with clearly defined right and wrong answers. However, asking such questions requires a controlled setting to ensure that no cheating occurs. This has become increasingly difficult due to the widespread use of technological devices and the omnipresence of internet search engines. Recent scholarship has demonstrated that due to respondent dishonesty, responses to factual knowledge

then the data is available upon request from Albert Weckman, Åbo Akademi University at albert. weckman@abo.fi (https://www.abo.fi/en/contact/ albert-weckman/).

**Funding:** LR received a grant as co-applicant for the Future of Democracy: Centre of Excellence, funded by the The Åbo Akademi University Foundation (https://stiftelsenabo.fi/en/home/). The funder has not played any role in the study design, data collection and analysis, decision to publish, or preparation of the manuscript.

**Competing interests:** The author has declared that no competing interests exist.

items in self-administered surveys cannot fully be trusted [3–6]. However, conducting face-to-face survey interviews is much more costly and time-consuming than, for example, web-based surveys. In panel designs, which may stretch across several decades and include numerous waves, face-to-face interviews are seldom a realistic alternative. In many occasions, this makes proper measurement of political knowledge impossible for many survey projects. For scholars, this is frustrating, because panel studies offer analytical possibilities that are far beyond cross-sectional, or even experimental, survey designs.

Secondly, with declining response rates causing much headache [e.g. 7], survey researchers are struggling to find ways to keep the hurdles for survey participation low. Asking factual knowledge questions risks respondent frustration. Political knowledge levels among the public are often low and no one likes to reveal their own ignorance by failing a knowledge quiz in a survey interview.

As online surveys are irresistibly becoming the dominant form for survey data collection, scholars have sought ways to deal with the problem of cheating. For example, Motta et al. [5] explore the impact of item wordings on the reliability of unsupervised political knowledge questions. Clifford and Jerit demonstrate that asking respondents to commit to honesty alleviates, but does not eliminate, the distortion caused by cheating [4] and Höhne et al. [6] and Marquis [8] develop techniques for identifying cheating behavior in web surveys (see also [9]). However, as Clifford and Jerit in particular demonstrate, cheating propensity varies across individuals and there is no waterproof way of identifying cheating on the individual-level [4], they suggest using (imperfect) aggregate-level measures to control for the effects of cheating.

So while tackling respondent dishonesty can make online measurement of political knowledge possible, it is an uncertain solution. This study explores another possibility–the use of a proxy measure. Political sophistication scholars have previously used a range of different proxies, such as self-assessed sophistication, education or interviewer assessments [10]. However, scholars have not properly analyzed the suitability of such proxies. One reason for this research lacuna is the lack of appropriate survey data with population-based samples that would include factual knowledge items, but also different potential proxies. Using a survey from Finland from 2020 with a sample representative of the voting age population, this study overcomes this problem. Additionally, another Finnish survey from 2008 data with an identical sample and identical measures for some of the proxies, allows replications of the key analyses. This allows the study to go beyond tests of validity and even assess whether the findings regarding the key proxies are consistent across two different measurements that are temporally quite far apart.

The main analysis of the 2020 data includes five potential proxies for political knowledge: self-assessed sophistication, political interest, internal political efficacy, accuracy of party placements on a left-right dimension and political participation. The study examines these potential proxies in terms of 1) *convergent validity*, that is, the extent to which they are measuring the same construct; 2) *criterion validity*, *the* extent to which the various proxies are associated with other variables that we would expect them to be associated with, based on what we know about political knowledge; and 3) *predictive validity*, *the* extent to which the proxies accurately predict political knowledge. Additionally, *face validity*, that is, the extent to which a variable on the surface appears to adequately measure what it is supposed to measure, is discussed but not tested.

The study finds that, overall, political interest is the best proxy for political knowledge. Unlike the other proxy candidates, political interest has high convergent validity, it demonstrates the expected associations with age, gender and education and is a stable predictor of political knowledge across the two genders and different levels of age and educational attainment.

## Survey measurement of political knowledge

The significance of being politically informed was widely recognized by political theorists already before the survey method was developed. Already during the early days of survey research in the US in the 1940s and the 1950s, questions of public competence in selecting leaders and forming meaningful political opinions became an active area of study [10]. However, it was not until a wave of studies beginning in the 1980s [e.g. 1, 2, 11–13] that scholars started to give the measurement of political knowledge proper attention. Since then, researchers have made several advances regarding the conceptualization and empirical measurement of political knowledge and its neighboring concepts.

In the empirical literature on political sophistication, many scholars have consistently found low levels of political knowledge among democratic publics. Consequently, also much of the methodological work regarding the measurement of knowledge has examined whether public ignorance about politics is a genuine finding or attributable to various shortcomings in measurement strategies. Scholars have, for example, looked at the impact of excessively inflexible coding methods [14] and the lack of incentives for respondents to try and provide correct answers [15]. Furthermore, traditional survey items may give a false impression of the range of knowledge ordinary citizens hold about politics. As Prior [16] has demonstrated, while people may not always be able to express verbally what they know about politics, they can, for example, recognize key political actors from pictures. Also question format has a significant impact on individual-level findings regarding who is knowledgeable and who is not [17] and there is a particularly persistent and vibrant scholarly exchange regarding the meaning of 'don't know' responses and guessing behavior. While some researchers suggest that there are systematic differences between respondents' tendency to say they 'don't know' [18], other analyses do not corroborate this finding [19]. Tsai and Lin [20] advance the idea that the measurement of knowledge is affected by guessing propensity and that the success of guessing depends on a person's level of sophistication. Taken together, research suggests that it is difficult to draw definite lines between knowledgeability and ignorance using simple, additive survey measures of political knowledge, which do not account for the nuances that lie underneath the surface of the actual responses.

Thanks to these and many other contributions, there is a good understanding among survey scholars about the problems, restrictions and best practices in the survey measurement of political knowledge. However, this methodological literature is relevant only in a context where political knowledge can be measured under controlled circumstances. Inspired by the possibilities offered by online surveys, a growing literature has started to examine unsupervised survey measurement of political knowledge. Although web-based surveys are cost-effective and convenient, the lack of surveillance by an interviewer is potentially a lethal problem for the reliable measurement of political knowledge, which can be distorted by cheating. With no interviewer or researcher present, checking correct answers online, from books or by asking someone else is not only possible but also probable.

Research has confirmed these concerns. When the opportunity presents itself, survey respondents routinely check the correct answers when they respond to political knowledge questions in web-based surveys. Cheating occurs even when respondents are explicitly asked to commit themselves to honesty [4]. Although mobile device use is associated with less cheating than computer use, looking up answers is a problem both in samples that consist of semi-professional respondents (such as those accessible through MTurk) and non-professional respondents in a probability sample [5]. Open-ended questions are usually more difficult than multiple-choice questions, and they seem to elicit even more outside searches for correct answers [6, 21].

Overall, the findings consistently show that cheating enhances political knowledge levels and that although the effects can be alleviated through the choice of question wording and appeals to the respondents' honesty, the basic dilemma remains. Moreover, existing research emphasizes that cheating in online surveys results in substantively different results regarding political knowledge compared to results from traditional surveys. Checking for correct answers muddles the observed effects; the resulting answers are no longer a measure of how well a person can recall political facts, but a measure of how well a person is able to perform the information search. As Smith, Clifford, and Jerit (2020) demonstrate, fact recall is a product of political interest, whereas ability to find information is related to the motivation to do well in the test. Consequently, they conclude that 'search engine use reduces the validity of political knowledge measures and undermines the ability to replicate canonical findings in the public opinion literature' [22].

Given these discouraging findings, scholars have suggested various remedies. Discouragement to use external information sources has some, albeit not enough, effect (e.g. [5]). Using visual instead of verbal knowledge items could offer a way forward [23], but research has yet to establish whether this would work as a replacement for traditional knowledge measurement.

Proxy measures offer another alternative. The use of proxies is in itself nothing new, as political knowledge has in previous research been measured through various indirect indicators. However, a comprehensive analysis involving several different candidates for a knowledge proxy has not been done. With online surveys becoming increasingly common and cheating is undeniably a major concern for the measurement of political knowledge, it is arguably timely to return to some of the most commonly used survey items and see if they might provide a feasible solution for survey scholars. The forthcoming analysis examines self-assessment of one's personal level of factual knowledge, political interest, internal political efficacy, party placements on the left-right dimension and political participation level as potential proxies for political knowledge. While the list is not exhaustive, it includes the most obvious proxies, which are also available in most standard surveys of political attitudes and behavior.

All of these proxy candidates, along with a commonly used measure of political knowledge, are available in a cross-sectional 2020 survey from Finland, which was conducted in face-to-face interviews. Additionally, identical measures for knowledge, self-assessment and political interest are available from a comparable survey from 2008, allowing a partial replication of the analysis through a repeated cross-sectional design.

## The proxy measures

### Self-assessment

Perhaps the most obvious candidate is self-assessment of one's own level of political sophistication. Various forms of self-assessments are widely used in surveys to measure, for example, ideological self-placement, political interest and personal health status. A typical survey measure of self-assessed political sophistication is a simple question such as 'how well informed would you say you are of political matters?'

Face validity is arguably very high for self-assessments of political sophistication, because the items explicitly refer to knowledgeability or use some other expression of expertise related to politics. Although a self-assessment would be a highly convenient way to measure actual political knowledge, cognitive biases could weaken its accuracy. According to the Dunning-Kruger-effect, which has received empirical support in many studies from different domains, low-performers are particularly poor at self-evaluating their own abilities [24]. This suggests that not only is there a risk that self-assessments are inaccurate, but their inaccuracy is likely to be different for those who truly are knowledgeable compared to those who are not. Moreover,

it is simply tempting for anyone to portray oneself as more politically sophisticated than what might actually be the case, because it is socially desirable to be politically informed. Hence, alongside, or in addition to, a possible Dunning-Kruger-effect, people might exaggerate their sophistication in surveys because they want to give a good impression. If, however, self-assessment turns out to be a suitable proxy, it offers a simple, one-item alternative that can be used conveniently in self-administered surveys.

## Political interest

Another apparent candidate is the self-declared interest in politics. It is available in practically all surveys about political attitudes and behaviors, which would also make political interest a very convenient proxy for political knowledge. The expression of political interest is widely considered to indicate motivation to engage with politics and which can be linked to behaviors such as following politics in the media and discussing it [e.g. 25]. If interest entails motivation, it seems plausible that interest would have a strong relationship with factual knowledge as well. People, who are interested in politics, or anything else, typically expose themselves to information about it.

While it is logically possible to be very interested in something but not be well informed about it (or vice versa), in real life the two are typically closely linked. In democracies, following politics is voluntary and therefore expressions of political interest are likely to reflect genuine motivation to become involved with political matters. However, self-expressed political interest could be particularly vulnerable to social desirability effects, because normative expectations regarding democratic citizenship tend to emphasize paying attention to politics as a citizen virtue. Although face validity seems high, it could be that respondents do not consider interest, i.e. an expression of motivation, as a matter of sophistication.

## Internal political efficacy

The two-dimensional concept of political efficacy includes an external and an internal component. External political efficacy refers to the feeling of system responsiveness to one's needs and wants, and its face validity as a proxy for knowledge seems poor. Internal political efficacy (IPE), on the other hand, involves a subjective evaluation of how well a person understands politics and feels capable of participating in it [see e.g. 26]. IPE is a widely used concept and it is routinely included in political surveys, in slightly varying formats.

IPE resembles self-assessment (see above), but it nevertheless taps into a slightly different aspect of self-evaluation. While a self-evaluation of sophistication requires the respondent to assess knowledgeability, IPE assesses ability to understand and take part in politics. IPE is measured using various combinations of (semi)standardized items, such as 'I consider myself to be well qualified to participate in politics' or 'Sometimes politics and government seem so complicated that a person like me can't really understand what's going on' [see e.g. 27]. Metaphorically, one could say that while a self-evaluation of sophistication asks whether the respondent is familiar with the parts of a car, IPE asks whether the respondent feels confident in driving a car. Despite this difference, face validity is likely to be high for IPE, as it includes a direct reference to personal ability to understand politics. However, given the variation in the specific items used to measure IPE, also face validity is likely to vary depending on these choices.

## Party placements

Another widely available and potential measurement is the placing of parties or their policy platforms on an ideological dimension, typically the left-right dimension. Used e.g. by [28] as a proxy for sophistication, comparing the respondents' party placements with those by experts, could be a close proxy for factual knowledge. The closer a person is to experts' party

placements, the more likely it is that the person is highly informed about politics. The underlying assumption is that more political knowledge leads to more accurate perceptions of party ideology, assuming that expert perceptions are accurate. A potential caveat is that respondents may be prone to evaluating preferred parties closer to their own ideological positions, rather than positioning parties according to an objective assessment. This pertains also to the face validity of party placements. Although many respondents are likely to think of party placements as a matter of factual knowledge in the sense that some placements are more 'correct' than others, the question itself is nevertheless framed as a matter of opinion, not as a matter of knowledge, as political knowledge items are. However, placing parties on any ideological continuum also depends on the country context. In Finland, where there are several parties with somewhat fluid ideological boundaries, there is always room for disagreement about their correct placement. In similar cases of high fragmentation in the party system, party placements are likely to be good proxies for knowledge. However, in two-party systems, where there is little discussion about which of the parties is, e.g., more conservative versus liberal, party placement is more likely a direct measure of basic knowledge about the system, rather than a proxy for it Consequently, usage of party placement as a knowledge measure is always context-dependent.

Although projections of own ideological positioning may cause some confusion in measurement, party placements show much promise as a potential proxy for factual knowledge items. However, non-responses, especially in contexts with a high number of parties, pose an analytical challenge. It is difficult to place certain niche parties on a left-right (or liberal-conservative) continuum. In those cases, choosing not to place a party at all could be a sign of sophistication, although in survey analysis such missing values are often interpreted as indicating ignorance. Moreover, a related problem is that it may not be sensible to ask people to place parties on a specific ideological dimension if that dimension lacks a connection to the party's political profile. Obviously, statistical imputation offers several alternatives for dealing with the resulting missing values, but all techniques also have limitations. In the forthcoming analysis, the missing values have been replaced by mean imputation, that is, by assigning the mean absolute difference between the respondents in the sample and the experts. The measure used in the analysis is the grand total of the absolute differences in evaluations of the eight parliamentary parties in Finland. Additionally, one MP in the Finnish parliament formed his own group alone. This one-person group was not included in the Chapel Hill survey and is therefore omitted. The values have been weighted for the number of parties that the respondent was able to place: the more parties a respondent placed on the ideological scale, the higher the (potential) value. For example: Respondent 1 places 8 parties on the scale, whereas Respondent 2 only places 6 parties. Assume that the grand total of the absolute differences between respondent and expert placement, is -10 for both respondents. For respondent 1, -10 is the total difference after 8 party placements whereas for respondent 2, the same score is the total difference after only 6 party placements. The weighted score for respondent 1 is calculated as follows: -10/8 = -1.25; and for respondent 2: -10/6 = -1.67. Thus, respondent 1 gets a score closer to 0, which indicates a better score than the score for respondent 2 who only placed six parties. For the analysis, the scores have been converted and rescaled into values between 0 and 1. By applying this weight, it is assumed that willingness to provide party placements in a multiparty context is in itself a sign of sophistication, whereas providing missing values is the result of inability to place parties.

## Political participation

Perhaps the most significant insight that has emerged from the mainstream empirical literature on political sophistication is that people who know more about politics also participate

more actively [1, 29, 30]. This finding is firmly anchored in the normative debate regarding the importance of sophistication for the functioning of democracy. It is widely accepted among the engaged scholars that political knowledge is a resource, which lowers the threshold for active participation and the realization of democratic citizenship [e.g. 1, 31].

The connection between participation and knowledge is therefore close both theoretically and empirically and e.g. Krosnick and Millburn [32] have used participation as an indicator for knowledge. Questions measuring (self-reported) political participation are also commonplace in surveys, which makes it another attractive option for a knowledge proxy. However, in comparison with the other proxy candidates, face validity seems low for participation. It is unlikely that questions asking about a person's political participation could be seen as questions intended to measure political sophistication. Consequently, the appropriateness of participation as a proxy for sophistication relies primarily on the empirical linkage between the two.

## Materials and methods

The analysis is primarily based on a survey conducted in Finland in 2020. The survey data used in the study has been gathered in accordance with the GDPR regulations. As the data does not include any sensitive personal data, an ethics committee approval was not required in Finland. Informed consent was acquired from each respondent verbally, as per the standard procedures of the survey company. Before asked to state if they would agree to being interviewed, the professional interviewers showed every respondent a written description of the survey, including the data protection statement. After that, respondents were asked whether they would consent to being interviewed. The consent was documented by the interviewer.

The 2020 survey replicated several political knowledge items and standard formulations of political interest and self-assessed sophistication from a 2008 survey. For these variables, the question wordings were identical in 2008 and 2020, and the same analyses will be repeated with both data for robustness. For the other variables (IPE, party placements and participation) only data from 2020 is available. The two surveys are also based on identical sampling of the voting age population. Both were conducted through face-to-face interviews by the same survey company and the same post-survey weighting method is used for both data. Both surveys were also conducted in election off-years, so that possible effects of a close surveillance of political campaigning among the public would not contaminate the findings. The expert data, used to calculate the deviations from expert party ratings, comes from the Finnish data in the Chapel Hill 2019 survey, available through https://www.chesdata.eu/2019-chapel-hill-expert-survey.

Table A1 in S1 Appendix reports all variable information, including original wordings and response categories. For the analyses, all variables have been standardized by using z scores.

## Results

Table 1 begins testing convergent validity through correlational analysis. The table shows the correlation coefficient and the 95% confidence intervals in brackets. All correlations are statistically significant (Spearman rank-order correlation, two-tailed significance at < .001-level).

All proxy candidates correlate at least moderately with knowledge and each other. This suggests that all the variables are inter-related, as expected. Self-assessment and political interest have the strongest correlation (.638). Knowledge correlates most strongly–but modestly–with political interest (.444) and almost as strongly with self-assessment (.431), and least with participation (.212). Consistent with these findings, in the 2008 data knowledge correlates with self-assessment and political interest at .450 and .408, respectively (Table A2 in S1 Appendix). This

**Table 1. Correlational analysis with the 2020 data (n = 1,097).**

|  | Knowledge | Self-assessment | Political interest | IPE | Party placement | Participation |
|---|---|---|---|---|---|---|
| *Knowledge* | - | .431 [.382 .478] | .444 [.395 .490] | .307 [.252 .360] | .372 [.319 .422] | .212 [.154 .268] |
| Self-assessment | .431 [.382 .478] | - | .638 [.601 .672] | .432 [.383 .479] | .267 [.210 .321] | .343 [.289 .395] |
| Political interest | .444 [.395 .490] | .638 [.601 .672] | - | .386 [.335 .436] | .259 [.203 .314] | .417 [.366 .465] |
| IPE | .307 [.252 .360] | .432 [.383 .479] | .386 [.335 .436] | - | .247 [.190 .303] | .273 [.217 .328] |
| Party placement | .372 [.319 .422] | .267 [.210 .321] | .259 [.203 .314] | .247 [.190 .303] | - | .242 [.185 .299] |
| Participation | .212 [.154 .268] | .343 [.289 .395] | .417 [.366 .465] | .273 [.217 .328] | .242 [.185 .299] | - |

reinforces the initial finding that self-assessment and political interest have highest convergent validity. The relationship between knowledge and interest is stable across measurement during off-election years and in the context of a general election. While the data used in this study comes from an off-election year, the latest Finnish parliamentary election survey data from 2019 (FNES 2019) was collected immediately after the elections. It uses a similar sample, a comparable 5-item political knowledge measure and an identical political interest measure. The Spearman correlation between knowledge and interest in FNES 2019 is .417, which is nearly identical to the .444 correlation reported in this analysis (Table 1).

Principal component analysis (PCA) provides a more durable test of convergent validity. The variables load on a single dimension (Table 2), suggesting that they measure a single construct. Horn's parallel analysis, which is the preferred method for ascertaining the appropriate number of factors retained by PCA [see e.g. 33], confirms the single-factor interpretation (Fig A1 in S1 Appendix). In other words, all proxies show potential as they all tap into the same underlying dimension.

Factor loadings and uniqueness reflect the strength with which each component is connected to the common dimension. As with the correlational analysis, party placement shows the weakest connection with the rest (weak loading, high uniqueness), while particularly self-assessment and political interest are the components, which are most closely connected to the common dimension.

Using data from 2008, Table 3 confirms the same pattern. For the three variables that are available in both data, the results are strikingly similar for both surveys. The variables load on a single dimension, and again self-assessment and political interest are more closely connected to the underlying dimension than knowledge (stronger loadings, lower uniqueness). This suggests that the underlying construct, which could be termed 'self-expressed cognitive engagement with politics', is more about cognitive engagement expressed through self-assessed sophistication, interest and a sense of efficacy, and somewhat less about objectively measured knowledge. The common dimension is also less connected to participation and party

**Table 2. Knowledge and its proxies: Principal component analysis with 2020 data (n = 1,097).**

|  | Loadings | Uniqueness |
|---|---|---|
| Knowledge | .562 | .652 |
| Self-assessment | .777 | .380 |
| Political interest | .817 | .322 |
| IPE | .567 | .678 |
| Party placement | .359 | .818 |
| Participation | .512 | .705 |

Note: Eigenvalue: 2.298, Chi$^2$ test $<$***

**Table 3. Knowledge, self-assessment and political interest: Principal component analysis with 2008 and 2020 data (n = 1,020/n = 1,097).**

|  | Loadings | | Uniqueness | |
| --- | --- | --- | --- | --- |
|  | 2008 | 2020 | 2008 | 2020 |
| Knowledge | .589 | .574 | .653 | .671 |
| Self-assessment | .746 | .788 | .444 | .378 |
| Political interest | .715 | .808 | .489 | .347 |

Note: 2008: Eigenvalue: 1.4141, Chi$^2$ test $<$***. 2020: Eigenvalue: 1.6047, Chi$^2$ test $<$***

placement. Hence, in terms of convergent validity, self-assessment and political interest stand out as the primary contenders for best proxy for political knowledge, based on both a correlational and a principal component analysis.

Criterion validity evaluates the extent to which the various proxies are associated with other variables as expected. As a first step, multivariate linear regressions were run, where the knowledge measure and the proxy candidates from the 2020 data take turns as the dependent variable, while gender, age and education are entered as independent variables in each analysis. The reasons for choosing (only) these variables are twofold. Firstly, canonical findings from previous research show that these variables are the most significant individual-level sociodemographic predictors of political knowledge. That men tend to score higher in survey knowledge questions is a persistent finding [e.g. 34]. Age typically shows a positive association with knowledge [35], while high education is a particularly strong predictor of high political knowledge [1]. In many studies, the association is so strong that education could also have been considered as another potential proxy, rather than a control variable. The bivariate correlations in the data used in this analysis were, however, lower for education than those reported above in Table 1, suggesting that education does not quite reach the same proximity to political knowledge as the other proxy candidates. In the 2020 data, Pearson correlations for education and knowledge and the proxies range between .226 and .361 and in the 2008 data between .227 and .304. Secondly, gender, age and education are available in practically all surveys that measure political attitudes and behavior. To maximize the generalizability of the findings across contexts, age, gender and education form the most appropriate set of independent variables for the current analysis.

Fig 1 below summarizes the results from a series of analyses (see Supporting information for full results and Fig A2 in S1 Appendix for the partial replication with 2008 data). It shows the coefficients from linear regressions with the 2020 data, where gender, education and age are used as predictors for political knowledge and the proxy candidates. Each analysis has been run separately, with gender, education and age entered simultaneously as independent variables, and knowledge and its proxies as the dependent variable, one at a time. All variables have been standardized using z-scores for comparability.

For knowledge, political interest and self-assessment, the observed patterns roughly follow expectations. Most significantly, male gender, education and age are strongly and positively associated with knowledge, which aligns with previous research. Additionally, interest and self-assessment show similar associations, although regarding age, the large CIs imply more individual variation than in the case of knowledge. For the others, the findings deviate substantially from knowledge. Age is negatively associated with IPE, party placement and participation, which conflicts with how they associate with knowledge. This suggests that using these variables as a proxy for knowledge might lead to substantially different findings.

Lastly, let us assess predictive validity, which, in psychometric testing, refers to the ability of a score on a scale to predict the values of a criterion measure. Here, the proxies represent the

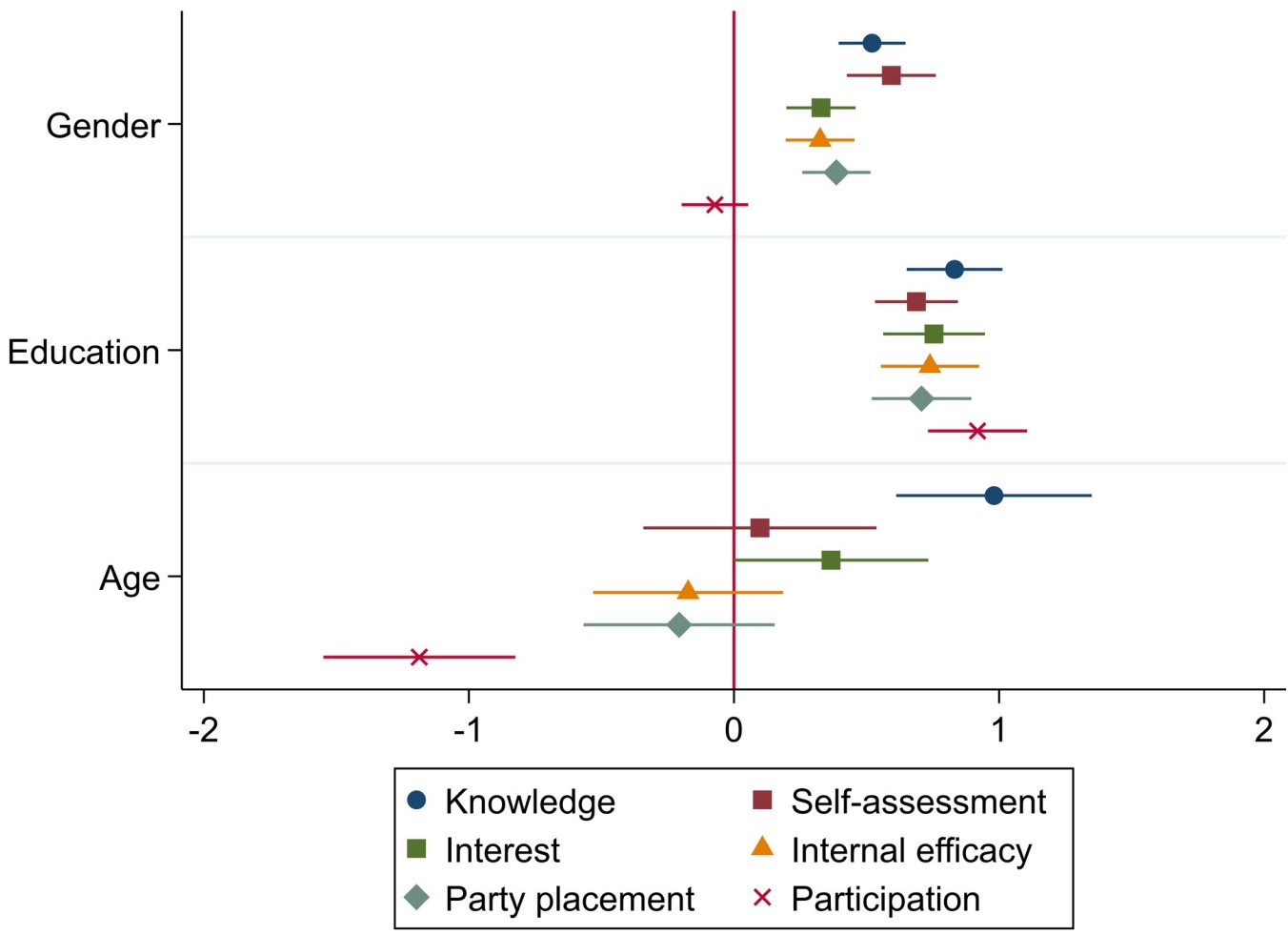

**Fig 1. Gender, age and education as predictors of knowledge and its proxies (n = 1,097, 95% CIs).**

scores and political knowledge is the criterion measure. Fig 2 compares the coefficients from five separate analyses, where one proxy candidate at a time and the three control variables have been entered as independents and where political knowledge is entered as the dependent variable. The figure only displays the coefficients for the proxies, and excludes the controls, for convenient reading (see Supporting information for full results and Fig A3 in S1 Appendix for partial replication with 2008 data). All variables have been standardized using z scores for comparability across the different measures.

All proxy candidates are statistically significant predictors of knowledge, when controlling for age, gender and education. However, self-assessment barely crosses the threshold of significance (p = .046), while all others are significant at p < .001. Comparing the magnitude of the coefficients, the ability to provide party placements that are similar to placements by experts stands out as the prime candidate in this comparison. Political interest is not far behind and the two have partly overlapping confidence intervals. Moreover, the range for CIs is significantly greater for party placements than for interest, suggesting that the latter might be less affected by individual-level differences.

Instead of using one of the proxies, they could all be combined into one factor, as suggested by the one-dimensional solution of the PCA reported in Table 2. All of the proxy candidates

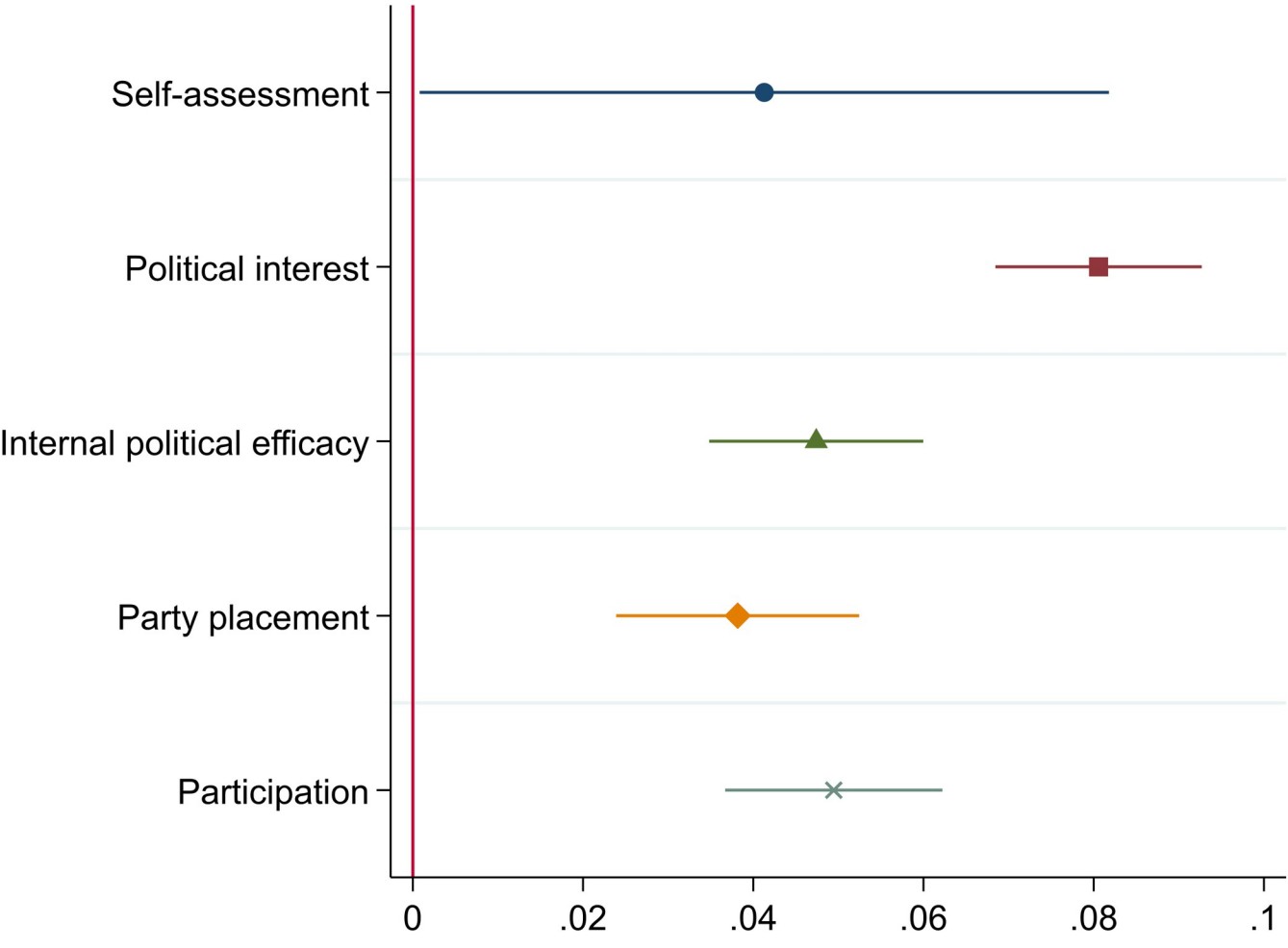

**Fig 2. The proxies as predictors of knowledge (coefficients and 95% CIs).**

might not be available in all surveys, which makes this solution less likely to be practicable, but nevertheless worth exploring. As reported in S4 Table in S1 File, the factor score for the proxy candidates (PCA, Bartlett method for estimating factor scores) is almost as strong a predictor of political knowledge as political interest. It is therefore also a viable method for using a proxy for knowledge, but in terms of predictive validity, using only political interest is at least as good a solution.

Another way to compare the magnitude of the coefficients is to enter all of them and the control variables in the same model and as a post-test compare the standardized coefficients. Table 4 reports the pairwise comparisons across all the proxy candidates based on such a regression model (detailed results and replication with the 2008 data in Tables A3 and A4 in S1 Appendix) to test whether the differences between the proxy coefficients are statistically significant.

The coefficients in Table 4 are based on z-scores, making the proxy candidates comparable in terms of how strongly they predict political knowledge. By comparing the F-values, it is possible to assess the probability of statistically significant differences between the predictive strength of the proxies. Again, political interest stands out as the prime candidate. It has a larger coefficient than the other candidates, suggesting a stronger association with political

**Table 4. Comparisons of coefficient magnitudes with the 2020 data.**

| Variable [coefficient] | Comparison (F) |
|---|---|
| Self-assessment [.013]–political interest [.058] | 18.78*** |
| Self-assessment [.013]–IPE [.016] | .13 |
| Self-assessment [.013]–party placement [.023] | 1.54 |
| Self-assessment [.013]–participation [.007] | .50 |
| Political interest [.058]–IPE [.016] | 18.33*** |
| Political interest [.058]–party placement [.023] | 11.83*** |
| Political interest [.058]–participation [.007] | 18.66*** |
| IPE [.016]–party placement [.023] | .64 |
| IPE [.016]–participation [.007] | .97 |
| Party placement [.023]–participation [.007] | 2.80 |

knowledge, and in comparison with the other candidates, the difference is statistically significant.

As a final test, let us examine whether the predictive ability of the proxy candidates varies across different categories of the control variables. In other words, do the proxy candidates have equal predictive validity for men and women, people in different ages and levels of educational attainment? Figs 3–7 below display the adjusted predictions for all combinations of the proxy candidates and the sociodemographic controls, in altogether 15 separate analyses with the 2020 data (see Supporting information for full results). In these analyses, statistically significant interaction coefficients are interpreted as suggestive evidence of poor predictive validity, as these suggest differences in how the proxies predict political knowledge across the sociodemographic controls. These differences are, however, sensitive to the choice of reference group. In the reported analyses, the group with supposedly lowest level of political knowledge was selected as the reference group (18-30-year-olds, female, comprehensive or less education).

Generally, the proxies predict knowledge similarly across the various respondent categories, with some significant exceptions. For political interest, there are no statistically significant interaction effects for age groups or education, but there is a gender difference. Predictive validity is higher for men, indicated by the steeper slope of the coefficient, which suggests that interest is more accurate in predicting political knowledge among men than women. A similar effect for gender is found for party placements. For party placements, predictive validity also varies across age groups although the differences are not substantial. For self-assessment, there are statistically significant interactions for age and education, suggesting that predictive validity for self-assessment varies across these groups. In the case of party placements, predictive

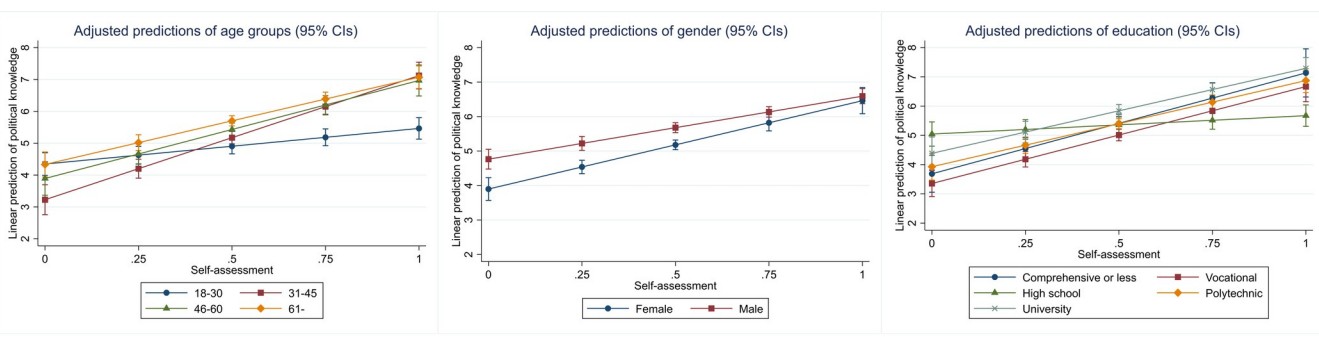

**Fig 3. Interactions between self-assessment and the sociodemographic variables.**

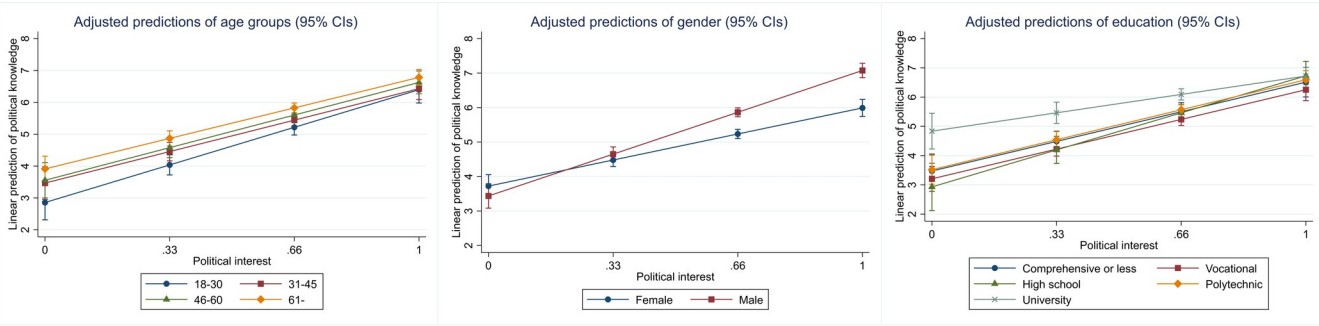

**Fig 4. Interactions between political interest and the sociodemographic variables.**

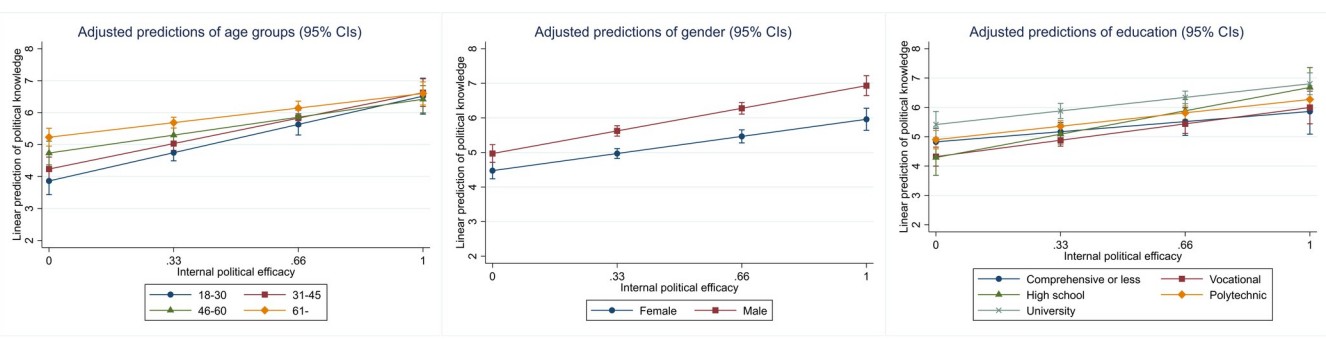

**Fig 5. Interactions between internal political efficacy and the sociodemographic variables.**

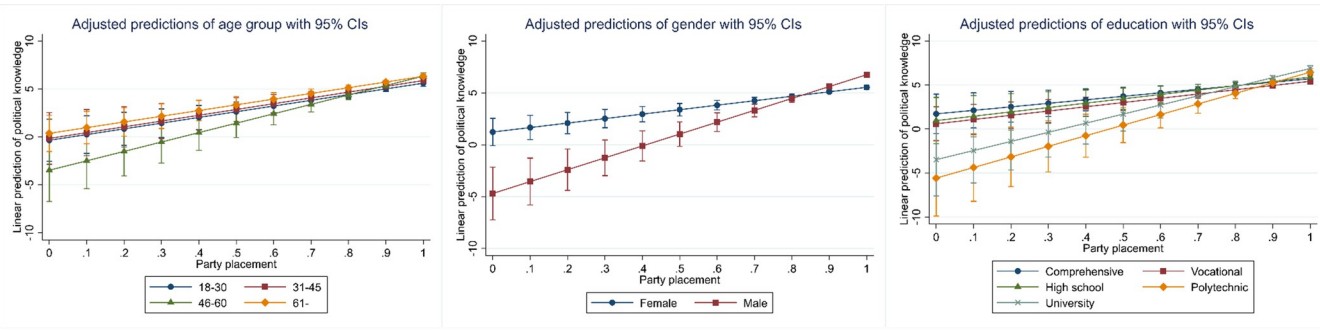

**Fig 6. Interactions between party placement and the sociodemographic variables.**

validity is highest for the youngest age group, 18 to 30-year-olds, who also differ statistically significantly from all the other groups. For education and gender, predictive validity is stable for party placements. When it comes to IPE, the only significant interaction is with age, suggesting that in this regard, IPE is a reasonably good predictor of knowledge.

Table 5 below offers a simple summary of all the preceding tests. The plus-signs indicate an assessment that a proxy candidate 'passed the test', whereas the brackets indicate borderline findings.

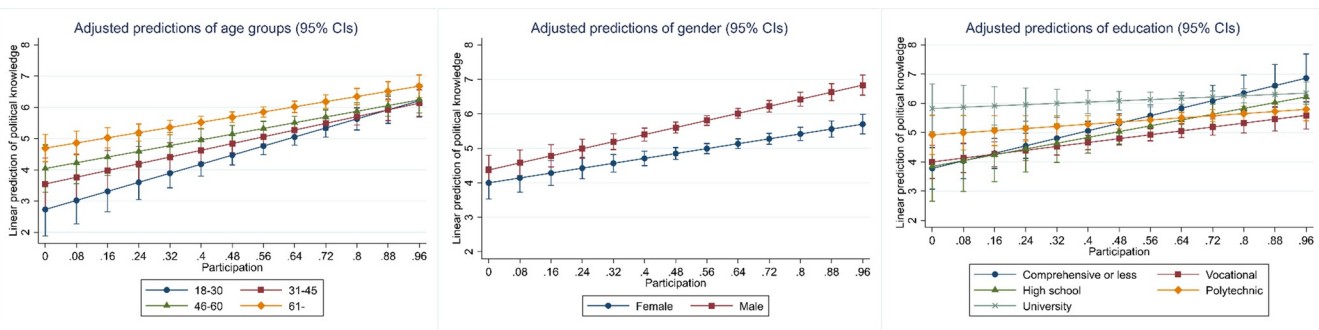

**Fig 7. Interactions between political participation and the sociodemographic variables.**

In terms of convergent validity, all proxy candidates showed some potential. They tap into the same underlying construct and show at least a moderately strong correlation with political knowledge. However, self-assessment and political interest are more closely correlated with knowledge and they also load more strongly to the common factor, suggesting they converge more intensely with knowledge than the rest.

For criterion validity, the observations are more straightforward. Strictly speaking, only political interest shows similar associations with age, gender and education, as political knowledge. Self-assessment comes close, but the large confidence intervals in the coefficient for age suggest that there is plenty of individual variation in terms of how accurately self-assessment predicts political knowledge. Age is a statistically insignificant predictor of IPE and party placements and a strongly negative predictor of participation. These observations clearly deviate from the expected pattern.

Political interest is the most consistent performer even in terms of predictive validity. Although the ability to correctly place parties on the left-right dimension has an even stronger predictive capability (controlling for age, gender and education) than political interest, the latter predicts political knowledge more evenly across age groups. IPE has also high predictive validity, but does not quite reach the same level as interest. In a summarizing assessment, political interest seems slightly better than party placements when it comes to predictive validity. Consequently, political interest is, among the candidates covered by the analysis, the most suitable proxy for political knowledge in terms of all three types of validity.

## Discussion

Survey research is struggling with low response rates. Online surveys continue to gain ground as a cost-efficient and respondent-friendly way to attract respondents. For reliable and valid measurement of political knowledge, this development presents a problem because of widespread respondent dishonesty in self-administered web surveys. This study has explored the

**Table 5. Summary of findings.**

|  | Convergent validity | Criterion validity | Predictive validity |
|---|---|---|---|
| Self-assessment | + | (+) | - |
| Political interest | + | + | + |
| Internal political efficacy | (+) | - | (+) |
| Party placement | (+) | (+) | + |
| Participation | (+) | - | - |

possibility of circumventing the problem through survey proxies. The analysis evaluated self-assessed political sophistication, political interest, internal political efficacy, placement of parties on an ideological dimension and political participation as potential proxies.

While all the proxy candidates show some promise, the recommendation that emerges from the analysis is that political interest is the best survey proxy for political knowledge. For survey researchers, self-reported political interest is an attractive option for indirect measurement of political knowledge, because it is simple and convenient. Particularly self-assessment, IPE and party placements are also likely to produce somewhat similar results as a typical index measure of political knowledge. However, in this analysis, none of them demonstrated the expected associations with the most widely used predictors of political knowledge, suggesting that each of them are partly driven by other factors than those that affect factual knowledge. In the typical situation, where researchers wish to make conclusions about the individual-level precursors of political knowledge, using these variables as proxies therefore runs the risk of misjudgment. If, on the other hand, only a rougher population-level estimate of political knowledge levels is needed, self-assessment, IPE and party placements are likely to be adequate.

So what would substituting knowledge for interest entail? The two are evidently related empirically, but do they tap into the same phenomena in the minds of survey respondents? Previous research gives reason to optimism. It is widely thought that an expression of political interest reflects a person's level of motivation to engage with politics. This understanding originates from psychology, where interest as a general concept is connected to feelings of motivation [36, 37]. Similarly, motivation has been considered also as a precursor of political knowledge [1]. It seems plausible that motivation is in some sense a shared 'root cause' for the empirical similarities between interest and knowledge, as demonstrated in this study. As Prior and Lupia [15], for example, have shown, the number of correct answers to knowledge questions increases with respondent motivation, which can be manipulated through rewards. Therefore, getting knowledge questions right in a survey interview is to some extent a product of motivational factors, such as interest.

However, responding to factual knowledge questions in a survey setting inevitably involves also a dimension of performance. Getting factual questions right requires an ability to perform well in a test situation, and some are better at it than others. Providing a self-assessment of political interest does not require a similar ability to perform in a test situation, but it could be vulnerable to social desirability bias, because declaring interest in politics is usually considered a citizen virtue. Therefore, using political interest as a proxy for political knowledge could mean substituting a problem with disentangling the measurement of knowledge from performance ability with a social desirability problem.

Although self-assessment and IPE fell short of interest in the overall empirical evaluation, they might nevertheless be acceptable proxies for knowledge as well, with some reservations. Both have arguably very high face validity, which could help in circumventing the problem of motivation versus test performance measurement, as discussed in regard to political interest. It seems plausible that self-assessment and IPE elicit responses reflect an informational, rather than a motivational component in the minds of survey respondents. Hence, the choice of appropriate proxy for political knowledge could to some extent depend on analytical context and design. Moreover, it is important to note that using political interest (or any of the others) as a proxy for political sophistication does not allow distinguishing between different dimensions of political sophistication [see e.g. 38, 39] or examining in detail how sophistication relates to other similar constructs. Based on this study, it is only possible to suggest that political interest is slightly better than other closely related survey items as a substitute for direct measurement of political knowledge. It is best employed in situations where a simple measure

of political knowledge is used as an explanatory variable or even as the dependent variable, but not in situations that involve a more complex conceptual design around the broader notion of political sophistication.

The conclusions in this study are based on a survey of a sociodemographically representative sample of the Finnish voting-age population from 2020, and partial replications using identical variables and sample from 2008. These data include an unusually rich set of political knowledge items and potential proxies. Although the analyses could only be replicated with self-assessment and political interest, the results were remarkably consistent in the two datasets across all the analyses. While this consistency gives confidence to the presented observations and suggests high test-retest reliability, the findings come with certain limitations and gaps remain for subsequent research. Some potential proxies for political knowledge were unavailable in the data, most notably a suitable measure of media consumption. The measure that was included in the survey asked how important the various forms of media are for the respondent for keeping informed about political matters. Although media preference and political news consumption are undoubtedly associated with political knowledge, the item was excluded from the analysis, because it is formulated in a way that is a bit unusual for most surveys. In election studies, for example, media consumption is typically measured by asking how much time a person has spent following the on-going or recent election campaign. The item that was available in the data focused on the preferred type of media, rather than intensity. To maximize generalizability of the results to typical survey designs across national contexts, only those proxies were included, which were measured with commonly used question wordings. For the same reason, only age, gender and education were included for testing criterion validity. However, subsequent research should include other variables, such as income. Further research is also needed to unlock regional differences across national contexts in terms of political knowledge, which could not be accounted for in this single-country analysis.

Moreover, future scholarship should design online surveys that make it possible to compare whether knowledge proxies and items asking to commit to honesty produce similar findings. This would help bridge the gap between the previous literature looking at honesty commitment items and the current study, which examines the use of proxies. In this way, we might reach a better understanding of whether one method of indirect measurement of political sophistication is somehow superior compared with the other. A related, but much broader, question is whether recruiting people to online versus offline surveys produce different kinds of samples. If so, this could even affect the validity of any sophistication measure, depending on whether it is employed in an online or offline survey. Although recent scholarship suggests different modes of recruitment lead to similar samples [40], more research into the matter is still needed.

## Supporting information

**S1 File.**
(DOCX)

**S1 Appendix.**
(DOCX)

## Acknowledgments

The author wishes to thank the editor, the reviewers, Prof. Mikko Mattila and Dr. Achillefs Papageorgiou for excellent comments, all of which significantly improved this article.

## Author Contributions

**Conceptualization:** Lauri Rapeli.

**Data curation:** Lauri Rapeli.

**Formal analysis:** Lauri Rapeli.

**Investigation:** Lauri Rapeli.

**Methodology:** Lauri Rapeli.

**Writing – original draft:** Lauri Rapeli.

**Writing – review & editing:** Lauri Rapeli.

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
