## [Decision Letter · Decision Letter 0]

9 May 2022

PONE-D-22-05474What is the best proxy for political knowledge in surveys?PLOS ONE

Dear Dr. Rapeli,

Thank you for submitting your manuscript to PLOS ONE. After careful consideration, we feel that it has merit but does not fully meet PLOS ONE’s publication criteria as it currently stands. Therefore, we invite you to submit a revised version of the manuscript that addresses the points raised during the review process.

We look forward to receiving your revised manuscript.

Kind regards,

Sean Richey

Academic Editor

PLOS ONE

Journal Requirements: 

a) Did participants provide their written or verbal informed consent to participate in this study?

b) If consent was verbal, please explain i) why written consent was not obtained, ii) how you documented participant consent, and iii) whether the ethics committees/IRB approved this consent procedure

Reviewers' comments:

Reviewer's Responses to Questions

**Comments to the Author**

1. Is the manuscript technically sound, and do the data support the conclusions?

Reviewer #1: Yes

Reviewer #2: Partly

2. Has the statistical analysis been performed appropriately and rigorously? 

Reviewer #1: Yes

Reviewer #2: Yes

3. Have the authors made all data underlying the findings in their manuscript fully available?

Reviewer #1: Yes

Reviewer #2: Yes

4. Is the manuscript presented in an intelligible fashion and written in standard English?

Reviewer #1: Yes

Reviewer #2: Yes

5. Review Comments to the Author

Reviewer #1: This manuscript is on an important topic, is clearly structured and written, uses appropriate data and methods, and produces interpretable results. My major hesitation is whether the findings are of sufficient interest and move forward our understanding enough to warrant publication. The problem of measuring political knowledge (either because there are no direct knowledge questions on a survey or because of concerns about reliability/validity of knowledge questions in online surveys) is a real one but not a new one. Researchers regularly use the kinds of proxies included in this manuscript. The analyses here provide evidence that political interest does a marginally better job than the other proxies (though none of them, to my mind, stand out as particularly equivalent to directly measuring knowledge. In addition, using proxies necessarily means one can’t study the various causal and interactional relationships among the various components of political sophistication if some of these components are used as stand-ins for the others. So what have we learned? If one wants to measure knowledge but does not have (or fears the reliability and validity of) direct knowledge questions, and one is not interested in the relationships among the components of political sophistication and/or the causes or effects of these distinct qualities, then using political interest as a substitute for knowledge is marginally the best choice. Is this enough of a contribution to be published in Plos One? I’m torn on this, though lean slightly to saying yes.

Specific Comments:

1. Abstract: Clear and concise on the manuscripts purpose, relevance to existing research, data, methods, and conclusions.

2. Introduction (pp. 2-5): Generally clear and informative, setting the stage for the research to follow. To my knowledge there is little evidence of respondent frustration or survey incompletion due to asking properly asked knowledge questions (p. 3, lines 51-54). The Clifford and Jerit research cited (p. 4, lines 58-59) would seem to address the problem being addressed in this manuscript – say more about why “another possible solution” is still needed?

3. “Survey Measurement of Political Knowledge” section (pp. 5-8): Summary of research on measurement and its potential shortcomings (pp. 5-6, lines 91-119) are generally accurate, though might specifically note the relationship between guessing and gender, which has been a major area of study/debate, as well as the substantive topics queried about. Stating that “Cheating occurs even when respondents are explicitly asked to commit themselves to honesty” (p. 7, lines 132-133) seems to contradict what is said in the prior section (P. 4, lines 58-59) and below (p. 7, lines 140-141. More clarity is needed. Nonetheless, the general point – that cheating occurs and that it reduces the validity of the measure – is an important one that is generally supported. The concluding argument regarding the use of proxies and the absence of research on how well they do is clear and convincing.

4. “Proxy Measures” section (pp. 8-13): Generally good overview of the proxies to be tested and the logic underlying the choices. I would have added “political attention” (i.e., self-reports on how often one follows politics) to the list. Also, party placement” is actually a (simple and limited) knowledge question and not really a proxy, despite the authors attempt to deny this (P. 11, lines 245-247. The approach to measuring party placement (pp. 12-13. Lines 248-275) is interesting, but not necessarily the obvious or only choice. I like the speculation on non-response as evidence of sophistication as opposed to ignorance, but this is just speculation. And the issue of what dimensions to place parties on seems an issue of question design and wording – we ae not locked into a single question on a single dimension. Nonetheless, the authors approach (imputation and weighting) is not an unreasonable one. In the end I remain concerned that the use of proxies conflate different components of citizenship that ideally we would want to keep distinct for purposes of identifying causal relationships among them, but this does not diminish the value of the research presented here.

5. “Materials and Methods” section (pp. 13-25): Could use more info about the surveys here (p.14, lines 295-307) -- e,g., who did the survey, sampling method and size, response rates, demographics relative to population, etc.- here or in an appendix. Would also be useful to know how the knowledge scale was constructed and its reliability. That the knowledge and proxy variables correlate is not surprising, To put the size of these correlations in perspective, the strongest correlations with knowledge (self-assessment and interest) “explain” roughly 20% of the variance in knowledge. The factor analysis does suggest a single dimension (something like “political sophistication”), though again this is not surprising. Also I’m not clear on how the factor analyses support the conclusion that self-assessment and interest are “the best proxies” for political knowledge (as opposed to best proxies for political sophistication). The convergent and predictive validity tests are interesting and provide some new/useful information regarding the use of specific proxies. I would have included partisanship and income in these analyses. The central conclusion drawn from these analyses – that relatively speaking, political interest appears to be the strongest proxy – is generally supported. I wonder if the authors thought of creating a measure combining the various proxies (perhaps based on the factor analyses, but excluding knowledge) to see how it performs relative to any single measure? Also, any reason to think the reliability of the various proxies (relative to each other and the knowledge measures) vary in important ways? Also any reason to think that responses to the proxies might also be affected/different in online surveys?

6. “Conclusion” section (pp. 25-27): Generally fair summary of findings, limitations, and implications.

Reviewer #2: The manuscript offers an examination of predictors of political knowledge based on data from two face-to-face surveys of voting-age Finnish residents. Self-reported knowledge, political interest, internal efficacy, party ideology assessment, and political participation are examined as potential proxies of factual political knowledge.

The paper is clearly written and presents solid arguments for the problematic nature of political knowledge metrics in online surveys. Two aspects of the data used in the study make it more valuable and interesting in my assessment. One is the national context – Finland is an interesting case since much of the political knowledge literature is US-centric. The second aspect is the face-to-face interviews used for data collection, a fairly rare modality these days.

The main challenges I see with the study are a certain lack of novelty, as well as lack of evidence that the examined variables are useful as proxy measures in practice.

In this work a series of variables are framed in the context of finding proxies for political knowledge. Those variables are not new -- there is already a vast body of literature informing our understanding of the relationships they have with political knowledge. The manuscript does not discuss very much of that existing literature when presenting the proxy variables, treating this as an exploratory study. Perhaps a case can be made that this is so in the Finnish context – overall though, the connections between all of those variables are well established.

The second challenge I see is that the study does not (and cannot, given its design) show that using these variables as proxies is really beneficial. To do that, we would have to confirm that those variables do a better job of representing the political knowledge construct compared to direct measurements of factual knowledge gathered through online surveys. Previous studies (e.g. Burnett, cited by the paper) give us some information by comparing the measurement of knowledge across survey modalities (online and offline). The present work confirms the known link between political interest and factual political knowledge. It also summarizes challenges that make online measurements of political knowledge somewhat problematic. Yet those two claims are not enough to show that the somewhat flawed online measurement of knowledge is a worse or more biased way of approximating political knowledge compared to using proxy variables that seem likely to be weaker as predictors.

A few other notes on the paper:

-- It would be helpful to give some standard information about the sample in the methods section – e.g. recruitment, incentives, demographics and their match with the general population demographics. One thing to consider/discuss more generally is if a sample of people that we can recruit for face-to-face interviews today skews in ways relevant to political knowledge compared to people who can be recruited to do online surveys.

-- One thing to note is that political interest and other variables may be more highly linked to knowledge in non-election years (when the surveys were done) when the media environment is less saturated with political information.

-- The study examines the connections of variables with political knowledge across gender, age, and education groups. One other variable that may be useful to include is income, unless it’s too highly correlated with education.

-- Some of the variables in the study may have a non-linear relationship with education, hence some of the patterns captured in the figures.

-- Ideally internal political efficacy would be measured through more than a single item.

-- The study suggests that using two datasets from different years can be used to examine test-retest reliability. That is likely not the case given that the two surveys had different participants. Moreover, political variables would not be expected to remain stable over a period of 12 years.

6. PLOS authors have the option to publish the peer review history of their article (what does this mean?). If published, this will include your full peer review and any attached files.

Reviewer #1: No

Reviewer #2: No

---

## [Author Response · Author response to Decision Letter 0]

29 Jun 2022

I thank the reviewers for a detailed reading of my manuscript and for all the insightful comments. I found the critical remarks to be fair and I have done my best to respond to them. My responses below roughly follow the order of the decision letter. I have provided responses to the reviewers’ comments and documented the actual changes I have made to the manuscript, for more convenient reading. I have also tried to format the manuscript according to the journal requirements. I look forward to hearing how you feel about the revised version.

1. Significance of the paper and its findings

Both reviewers challenge whether the analysis is interesting and relevant enough to warrant publication. Although the reviewers, thankfully, slightly lean towards an affirmative answer, it is still necessary to give the issue proper attention.

Firstly, as reviewer 1 points out, researchers regularly use those variables, which I examine in the paper, as proxies for knowledge. However, since there is almost no research into how those proxies actually relate to knowledge, this lacuna is arguably in itself a strong reason for why papers such as the one at hand are needed. My paper, if accepted, will not provide all answers, but it would serve as a useful guiding post for further research and it could, I hope, offer some concrete information to those survey researchers who are forced to resort to using a proxy for knowledge. Until now, they have had very little to go on when choosing between possible proxies. In my opinion, an important argument that supports publishing the results is (as I have tried to argue in the paper) that a further expansion of web-based surveys seems inevitable and the validity problem of knowledge questions will only grow in importance. We still do not have optimal solutions to deal with that problem. Any step towards a feasible solution is, in my view, valuable. So far, research has portrayed the use of honesty appeals in web surveys as to some degree useful but imperfect, but the use of proxies has not been properly explored. This is how I have sought to frame the paper, both in the Introduction and in the concluding Discussion. Additionally, thanks to reviewer comment #2 below, I now include a better explanation for why existing literature on alleviating online cheating is not enough.

However, I also understand the reviewer’s hesitation regarding the primary finding(s) of the paper. As the reviewer correctly concludes, it seems that political interest is slightly better than the other proxy candidates, but using any of them is problematic, if one wishes to investigate how knowledge associates with its close conceptual relatives or be able to distinguish between different dimensions of sophistication. I agree and admit that in the initial submission I had not adequately discussed under what circumstances and in what kinds of research designs the investigated proxies could actually be used – and when using them instead of direct measurement of knowledge is not a good option. To address this, I suggest adding the following paragraph to the Discussion:

Moreover, it is important to note that using political interest (or any of the others) as a proxy for political sophistication does not allow distinguishing between different dimensions of political sophistication (see e.g. (35,36)) or examining in detail how sophistication relates to other similar constructs. Based on this study, it is only possible to suggest that political interest is slightly better than other closely related survey items as a substitute for direct measurement of political knowledge. It is best employed in situations where a simple measure of political knowledge is used as an explanatory variable or even as the dependent variable, but not in situations that involve a more complex conceptual design around the broader notion of political sophistication.

Reviewer #2 also wonders about the novelty of the analysis, although from a slightly different perspective. What is new about this analysis, asks the reviewer, when there is already plenty of research that looks into the relationships between political knowledge and the proxies? It is true that there is research where, for example, some of the proxies are among the key independent variables used to explain political knowledge / sophistication. There is also some research, which focuses specifically on, for example, the relationship between (internal) efficacy and knowledge or on education and knowledge.

These studies do not, however, examine the possibility of using variables included in this study as proxies for political knowledge. Consequently, previous literature does not include research designs that would 1) include all five of these variables in a comparative setting, 2) in order to assess them in terms of various types of validity 3) with the explicit aim to determine which of them could be used as a survey proxy for knowledge. Whereas previous literature approaches the same relationships from the perspective of how they relate to patterns of political behavior or political cognition, the current manuscript has a methodological approach, which also utilizes the rare possibility to measure several closely related concepts at the same time. To underline the main point here - the current study offers the first examination where a number of plausible proxies are included in the same analysis and evaluated explicitly as proxies, rather than included in regression models as predictors, control variables or closely related concepts.

To me, as a (survey) scholar of political sophistication, the matter is significant enough to warrant publication (obviously), but I fully realize that this is debatable. I hope that the addition documented above and my response to the next comment have made the paper more convincing regarding this matter.

2. Connection to previous literature

Moreover, two additional comments by Reviewer #1 also touch upon the same issue of contribution and the positioning of this paper in existing literature. Firstly, the reviewer points out that since Clifford and Jerit already seem to suggest a solution to online cheating, some more explaining is needed as to why this analysis is needed. Secondly, the reviewer correctly notes that the reference to Clifford and Jerit on lines 58-59 is not entirely compatible with what is said later on p. 7.

The questions are fair and I thank the reviewer for locating these spots where I have been too imprecise in my recaps of Clifford and Jerit (whose study is an important predecessor for this submission).

When it comes to the first point, Clifford and Jerit conclude that asking respondents to commit to not cheating is the *best* way to tackle cheating *among the ones they studied*. However, they find that the tendency to cheat varies between different types of individuals and samples so they end up recommending using aggregate-level assessments of cheating rather than individual-level. What is maybe most important, they concede that “At present, researchers do not have a definitive way to identify cheating.” (p. 874) Consequently, I would argue that in addition to using verbal commitments to honesty in online surveys, we need to also look at other possibilities, because 1) verbal commitments are not bulletproof and their effect varies across contexts and 2) we can never be absolutely sure how widespread cheating really is. To better explain this in the manuscript, I suggest the following addition (see lines 61-64):

However, as Clifford and Jerit in particular demonstrate, cheating propensity varies across individuals and there is no waterproof way of identifying cheating on the individual-level (4), they suggest using (imperfect) aggregate-level measures to control for the effects of cheating.

And the following paragraph (lines 65-67) now continues by pointing out what the contribution of the current study is, given the uncertainties revealed by previous research:

So while tackling respondent dishonesty can make online measurement of political knowledge possible, it is an uncertain solution. This study explores another possibility– the use of a proxy measure.

As for the second point, this was just careless on my part. The original lines 58-59 oversimplified what Clifford and Jerit actually find. This could be easily fixed by rewording the original lines 58-59 as follows:

Clifford and Jerit demonstrate that asking respondents to commit to honesty alleviates, but does not eliminate, the distortion caused by cheating

Additionally, the text added to fix point #1 (see above) further clarifies what the study by Clifford and Jerit really concludes.

3. The proxies included / excluded 

Reviewer #1 makes valid comments about the list of proxies analyzed in the paper. I fully understand the reviewer’s comment that there is no variable measuring (media) attention to politics. Admittedly frustrating, the measure that is available only measures the choice of media for consumption of political news, not intensity. As I have explained in the concluding discussion, there is no proper and commonly used measure of political attention available in the data. While it could have been possible to do something with the sub-optimal item that is available, I chose not to, because it was not in line with the overall design of the study. I believe that much of the strength of the current study lies in the fact that the proxy items are essentially identical with those commonly used in surveys everywhere. This makes the analysis useful for a broad audience of survey researchers and my concern is that using one measure, which deviates from all others in this crucial aspect, would be harmful to the study as a whole. To acknowledge this, a paragraph in the discussion comments on the absence of a media attention measure and agrees with the reviewer that political attention would have been on the list had a proper measure been available.

The reviewer also makes valuable comments regarding party placement. I am fully aware that it is nowadays commonly accepted that there are other ideological dimensions in the (typical) political space in Western democracies besides left/right. In many countries, there is at least one other ideological dimension, a cultural dimension, which crosscuts the left/right and is therefore distinguishable from it. In this sense, only relying on left/right is, of course, insufficient. However, given the continued prominence of left/right as a dimension that structures democratic politics, it is handy for survey researchers, because it is familiar to most respondents and available in most surveys. Measure of the cultural dimension (or similar construct) are becoming more widely available, but unfortunately such a measure was not present in the data used here.

The reviewer, on the other hand, contests whether we should not in fact consider placing parties on an ideological left/right spectrum as a direct indicator of knowledge, rather than a proxy. This is a fair question. Given that it is possible to think that parties have a ‘correct’ placement on an ideological dimension, it is also possible to see the question as a direct test of knowledge.

While I readily admit that the party placement is very close to a direct knowledge measure, I would nevertheless defend considering it as a proxy for knowledge on two grounds. Firstly, in the case of multiparty contexts with some fluidity between parties’ ideological boundaries, not even experts agree to a 100 percent on the exact placement of all parties. While they will agree, also in the Finnish case, which parties are considered leftist and rightist, the evaluations are not identical. Secondly, in the survey(s), the question is not framed as a knowledge item, while the direct measurements of knowledge are. Party placement is framed as an expression of one’s view or understanding, not as a test of whether one knows where a party should be placed. Such a framing leads the respondent to reflect on the issue rather than look for an objectively correct response. Hence, respondents might express an opinion about where they think a party currently should be placed, because e.g. of a particular stand on a big, topical policy question, as opposed to where that same party historically, and in a more general assessment, should be placed. In other words, it seems likely that most respondents will approach this question as a matter of expressing a personal view or understanding, rather than as a matter of correct/incorrect. Additionally, as mentioned in the manuscript, respondents may even project their ideological self-identification on these responses.

That said, I understand that party placements could, especially in majoritarian two-party contexts, be considered as a question of knowledge and nothing else. In fact, this was mentioned in the original submission where party placement is introduced as a proxy candidate. However, to further stress the point, I have now expanded this discussion, which now reads as follows:

Although many respondents are likely to think of party placements as a matter of factual knowledge in the sense that some placements are more ‘correct’ than others, the question itself is nevertheless framed as a matter of opinion, not as a matter of knowledge, as political knowledge items are. However, placing parties on any ideological continuum also depends on the country context. In Finland, where there are several parties with somewhat fluid ideological boundaries, there is always room for disagreement about their correct placement. In similar cases of high fragmentation in the party system, party placements are likely to be good proxies for knowledge. However, in two-party systems, where there is little discussion about which of the parties is, e.g., more conservative versus liberal, party placement is more likely a direct measure of basic knowledge about the system, rather than a proxy for it. Consequently, usage of party placement as a knowledge measure is always context-dependent.

4. Information about the surveys

Both reviewers were hoping for more basic information about the surveys. This is a reasonable request and I have added the following information to the Appendix:

Data description

2008 data

The following data description is from the Finnish Social Science Data Archive, where the data is deposited and available. The data identification code is FSD2499:

Target population: Finnish citizens aged 18 or over living in Finland, excluding the Åland Islands

Data collector(s): Taloustutkimus

Mode of data collection: Face-to-face interview

Sampling procedure: Probability; Stratified Quota sampling based on age, gender and municipality of residence.

2020 data

The data is deposited in the Finnish Social Science Data Archive, from where it will be accessible for research.

Target population: Finnish citizens aged 18 or over living in Finland, excluding the Åland Islands

Data collector(s): Taloustutkimus

Mode of data collection: Face-to-face interview

Sampling procedure: Probability; Stratified Quota sampling based on age, gender and municipality of residence.

There is no official documentation about response rates, but based on in-person exchanges with the survey company, there were typically 10 contacts made for each f2f-interview. This is a typical rate for such surveys in Finland. Moreover, as explained in the manuscript, the slight demographic skews in both data were corrected using post-survey weights. The weights were constructed using official population statistics from Statistics Finland.

5. Comments about the variables and scales

Reviewer #1 wonders how the knowledge scale was constructed. The origins of the items date back to the planning of the 2008 data, which included approximately 40 political knowledge items. Those items were chosen roughly based on the same procedure used by Delli Carpini and Keeter (1996), who surveyed a large number of political scientist about what people should know about politics. The goal of the 2008 survey was to produce a comprehensive picture of what the Finnish electorate knows (for a description, unfortunately only in Finnish: Elo & Rapeli, 2008; see also Elo & Rapeli, 2010).

Based on the reasoning about the dimensions of political knowledge (see Elo & Rapeli, 2010; Barabas et al. 2014), the 2020 data repeats parts of the 2008 knowledge items. As the figures below demonstrate, the distributions across the samples are similar and the sample means are almost identical.

The knowledge scales, which have been calculated by simply adding up the correct answers, distinguish between individuals in terms of what they know about politics and the individual-level predictors of (high) knowledge are in both cases fully in line with canonical findings: male gender, age and education are all positively associated. To avoid harmful guessing behavior, ‘don’t know’ responses were not encouraged (Mondak, 2001).

The reviewer also commented that partisanship and income could have been useful to include. This is true and something I also considered very carefully when planning the analyses. Both are indeed likely to be related to knowledge and its proxies. My reasons for choosing not to initially include them, and for suggesting they also now remain outside analyses, are twofold. Firstly, I wanted to only include control variables that are more or less universal across surveys that are conducted across democratic publics. There is a good deal of variation in how income is operationalized, e.g. as personal income, annual household income or subjective feeling about how well one can cope with current (household) income. Although I nevertheless admit that including an indicator of income, even if full concordance with other surveys might be lacking, could have been reasonable, my second reason made me lean towards excluding it. As it now stands, the analysis is already quite extensive with a high number of figures etc. My main concern has been not to cram too much into one paper, but instead conduct proper analyses with the most essential variables. In my reading of the literature, and with reference to the point made above, I chose to go with age, gender and education as the most important and suitable control variables. However, I fully understand that the optimal solution here is debatable.

Additionally, the reviewer also asks whether combining the proxies through a factor score, instead of using them individually, would be feasible. Yes, this possibility occurred to me too. I originally left it out, because I was (again) worried about the scope of the analysis becoming too large and because all of the proxies are unlikely to be available in many surveys anyway. Hence, I thought, such an analysis would probably not be helpful to many readers.

However, as the reviewer also suggested it, I thought it would be a good idea to include the analysis. I ran a similar analysis of predictive validity as the one presented in Figure 2. It shows that a factor score of the proxies is almost identical in predictive strength as political interest, political interest being marginally stronger. I now refer to this analysis in the body text and include the table in the Supporting information in order not to expand the main text too much. Here is the text added to the manuscript:

Instead of using one of the proxies, they could all be combined into one factor, as suggested by the one-dimensional solution of the PCA reported in Table 2. All of the proxy candidates might not be available in all surveys, which makes this solution less likely to be practicable, but nevertheless worth exploring. As reported in Table S4 in the Supporting information, the factor score for the proxy candidates (PCA, Bartlett method for estimating factor scores) is almost as strong a predictor of political knowledge as political interest. It is therefore also a viable method for using a proxy for knowledge, but in terms of predictive validity, using only political interest is at least as good a solution.

The table added to Supporting information:

Table S4: Factor score1 of PCA with all proxies as the predictor of political knowledge

 Coef. SE t P>t 95% conf. interval

Factor score .0801825 .0076128 10.53 0.000 .0652442 .0951208

Gender (male = 1) .0687076 .0120261 5.71 0.000 .0451094 .0923058

Education .063963 .0192175 3.33 0.001 .0262533 .1016726

Age .1948574 .0346664 5.62 0.000 .1268332 .2628815

Constant .4384795 .0265519 16.51 0.000 .3863779 .490581

1 Factor score for single factor derived from PCA including all the potential proxy measures. The coefficient is the z-standardized factor score, to ensure comparability with the other analyses.

Observations = 1,043

F(4, 1038) = 72.71

Prob > F = 0.0000

Adj. R-squared = 0.2854

6. Lack of online/offline comparison

Reviewer #2 makes a good point about the fact that the study cannot assess whether the use of proxies instead of direct measurement in online surveys is somehow beneficial, because the two cannot be directly compared. This is, of course, true. While it is regrettable that such a comparison is not possible given the research design, I would nevertheless argue that the current design has other, unusual benefits. The two surveys provide, according to my knowledge, the only existing survey data that allows testing the relationships between a sound knowledge measure and a number of potential proxies. Hence, the focus of the paper has been on this particular advantage.

The reviewer is, however, correct in pointing out that we still do not know whether a proxy would work better in online settings than some type of an encouragement to refrain from cheating. I have added a paragraph discussing this and it reads as follows:

Moreover, future scholarship should design online surveys that make it possible to compare whether knowledge proxies and items asking to commit to honesty produce similar findings. This would help bridge the gap between the previous literature looking at honesty commitment items and the current study, which examines the use of proxies. In this way, we might reach a better understanding of whether one method of indirect measurement of political sophistication is somehow superior compared with the other.

Additionally, Reviewer #2 had some important, minor comments, which I have responded to as follows:

7. Differences in samples in online/offline surveys

The reviewer suggests discussing the more general issue of how potential differences in respondent recruitment to f2f versus online surveys may affect the measurement of political knowledge. Although this might perhaps be slightly outside the core message of the article, it is certainly an important aspect that deserves to be mentioned.

A recent study sheds some light into the matter. Cornesse et al. (2021) analyze whether different modes of recruiting to an online panel result in different samples in terms of key sociodemographic characteristics. The modes they study are online-only, concurrent mode, online-first, and paper-first. They conclude that all modes lead to samples that are similar when it comes to sociodemographic representation. Although this cannot be considered conclusive evidence, the findings by Cornesse et al. are cause for optimism that sociodemographic differences, which are the main drivers of sophistication, could in fact be equally accounted for in both online and offline survey settings.

I see this issue as an issue that should be noted as a possible limitation to the reported findings and therefore suggest adding the following to the Discussion:

A related, but much broader, question is whether recruiting people to online versus offline surveys produce different kinds of samples. If so, this could even affect the validity of any sophistication measure, depending on whether it is employed in an online or offline survey. Although recent scholarship suggests different modes of recruitment lead to similar samples (37), more research into the matter is still needed.

8. Political interest in non-election years and election years

An excellent point by reviewer 2. It is indeed possible that the connection between political interest and knowledge might look different depending on whether it is measured in a survey that is conducted in conjunction with an election (e.g. national election studies) or in a non-election year (such as in this case). Although this is speculation, I also find it plausible that people could, for example, report higher levels of political interest during an election or immediately after it than during an off-year. Moreover, as the reviewer suggests, when political information is widely available during election time through the media, many people are likely more attuned to politics and perhaps even be better informed.

To address this, I checked the Spearman correlation between a comparable 5-item political knowledge measure and an identical political interest measure in the latest Finnish parliamentary election survey data from 2019 (FNES 2019). It is a post-election survey, conducted immediately after the parliamentary elections, by the same survey company as the data used in this paper, using a similar sample and a similar weight to correct for sociodemographic imbalances. The FNES 2019 data is downloadable in English through https://services.fsd.tuni.fi/catalogue/FSD3467?lang=en&study_language=en.

The correlation between knowledge and interest in the FNES 2019 data is .417, whereas it is .444 in this analysis (Table 1). So the correlations are almost identical, when using similar variables + samples, regardless of whether the measurement is from an off-election year or from a study conducted in conjunction with an election. This gives me confidence in the key finding in the paper regarding the suitability of interest as a proxy for knowledge. To explain this in the paper, I have added the following footnote:

The relationship between knowledge and interest is stable across measurement during off-election years and in the context of a general election. While the data used in this study comes from an off-election year, the latest Finnish parliamentary election survey data from 2019 (FNES 2019) was collected immediately after the elections. It uses a similar sample, a comparable 5-item political knowledge measure and an identical political interest measure. The Spearman correlation between knowledge and interest in FNES 2019 is .417, which is nearly identical to the .444 correlation reported in this analysis (Table 1).

9. Non-linear relationship between education and the proxies

The reviewer notes that some of the patterns between education and the proxies are likely due to non-linear relationships. I can only agree – it is often the case that people with a university degree are particularly distinguishable from other educational groups.

10. Measurement of internal political efficacy

Yes, I fully agree that it would have been better to have more items that tap into the sense of internal political efficacy. Although nothing can be done at this stage, it may be worth noting that, on the positive side, the one item that is included is arguably the most relevant among the 2-4 standard items used to measure internal political efficacy and that the two surveys used identical wordings.

11. Test-retest reliability

The reviewer points out that since the respondents were not the same in the two surveys, test-retest reliability cannot be examined. The reviewer is right, this was a sloppy formulation. I suggest changing the original sentence to the following, which I believe more accurately states what I think is relevant here:

This allows the study to go beyond tests of validity and even assess whether the findings regarding the key proxies are consistent across two different measurements that are temporally quite far apart.

References

Barabas J, Jerit J, Pollock W, Rainey C. 2014. The Question(s) of Political Knowledge. American Political Science Review 108(4),840–855.

Delli Carpini M X, Keeter S. 1996. What Americans Know About Politics and Why It Matters. Yale University Press: New Haven, CT.

Elo, K, Rapeli, L. 2008. Suomalaisten politiikkatietämys [What Finns Know about Politics]. Edita Prima: Helsinki.

Elo, K, Rapeli, L. 2010. Determinants of Political Knowledge: The Effects of Media on Knowledge and Information. Journal of Elections, Public Opinion and Parties 20(1), 133 – 146. 

Mondak J. 2001. Developing Valid Knowledge Scales. American Journal of Political Science 45(1), 224-238.

Cornesse C, Felderer B, Fikel M, Krieger U, Blom AG. 2021. Recruiting a Probability-Based Online Panel via Postal Mail: Experimental Evidence. Social Science Computer Review (Online First). doi:10.1177/08944393211006059

---

## [Editor Report · Decision Letter 1]

21 Jul 2022

What is the best proxy for political knowledge in surveys?

PONE-D-22-05474R1

Dear Dr. Rapeli,

We’re pleased to inform you that your manuscript has been judged scientifically suitable for publication and will be formally accepted for publication once it meets all outstanding technical requirements.

Kind regards,

Sean Richey

Academic Editor

PLOS ONE
---

## [Editor Report · Acceptance letter]

11 Aug 2022

PONE-D-22-05474R1 

What is the best proxy for political knowledge in surveys? 

Dear Dr. Rapeli:

I'm pleased to inform you that your manuscript has been deemed suitable for publication in PLOS ONE. Congratulations! Your manuscript is now with our production department. 

Kind regards, 

on behalf of

Dr. Sean Eric Richey 

Academic Editor

PLOS ONE